# Preparation and Electrochemical Performance of Three-Dimensional Vertically Aligned Graphene by Unidirectional Freezing Method

**DOI:** 10.3390/molecules27020376

**Published:** 2022-01-08

**Authors:** Peng Xia, Zhenwang Zhang, Zhihong Tang, Yuhua Xue, Jing Li, Guangzhi Yang

**Affiliations:** School of Materials and Chemistry, University of Shanghai for Science and Technology, Shanghai 200093, China; xiapeng20211@163.com (P.X.); 183762634@st.usst.edu.cn (Z.Z.); zhtang@usst.edu.cn (Z.T.); xueyuhua@usst.edu.cn (Y.X.); Lijing6080@usst.edu.cn (J.L.)

**Keywords:** three-dimensional vertically aligned graphene, unidirectional freezing, electrochemical performance

## Abstract

Three-dimensional vertically aligned graphene (3DVAG) was prepared by a unidirectional freezing method, and its electrochemical performances were evaluated as electrode materials for zinc−ion hybrid supercapacitors (ZHSCs). The prepared 3DVAG has a vertically ordered channel structure with a diameter of about 20−30 μm and a length stretching about hundreds of microns. Compared with the random structure of reduced graphene oxide (3DrGO), the vertical structure of 3DVAG in a three−electrode system showed higher specific capacitance, faster ion diffusion, and better rate performance. The specific capacitance of 3DVAG reached 66.6 F·g^−1^ and the rate performance reached 92.2%. The constructed 3DVAG zinc−ion hybrid supercapacitor also showed excellent electrochemical performance. It showed good capacitance retention up to 94.6% after 3000 cycles at the current density of 2 A·g^−1^.

## 1. Introduction

The electrochemical energy storage and transfer devices of supercapacitors have the advantages of high power, fast charge and discharge speed, long life, and safe operation, showing great potential in portable electronic products, hybrid electric vehicles, implantable biomedical devices, uninterruptible power supplies, and grid energy storage [1,2,3]. Zinc-ion hybrid supercapacitors (ZHSCs) are regarded as greatly promising energy storage devices which have attracted much attention due to their high theoretical capacity, good electrical conductivity, and low redox potential [4,5,6]. In ZHSCs, the use of carbon materials as capacitor electrode materials is due to their advantages of large specific surface area, high porosity, high conductivity, good chemical stability, abundant reserves, nontoxicity, and being harmless [7,8]. Generally speaking, a larger specific surface area of the carbon material results in larger electric double-layer capacitance. Many researchers have reported that the pore structure, electrical conductivity, and surface properties of carbon materials also affect electrochemical performance [9]. Carbon materials used for ZHSC capacitive electrode materials mainly include activated carbon [10,11], porous carbon [12,13], nitrogen-doped tubular carbon [14], graphene [15,16], metal-organic framework−derived carbon, and mesoporous carbon hollow spheres [17]. Among these materials, graphene has shown excellent performance because of its unique structures and properties in mechanical strength, elasticity, carrier mobility, and electrical and thermal conductivity [18,19]. 

In recent years, three-dimensional vertically aligned graphene (3DVAG) has received more and more attention as an electrochemical electrode material due to its superior reaction kinetics and mass transfer ability. 3DVAG has a vertical open passage and low pore curvature, which can increase the load of active materials while ensuring efficient ion and electron transport, thereby improving the capacitance performance of the electrode, while the three-dimensional structure can also improve the mechanical stability of the electrode [20].

The preparation methods of 3DVAG mainly include the directional freezing method [21,22,23,24], plasma−enhanced chemical vapor deposition (CVD) method [25,26,27,28,29], and KOH−assisted hydrothermal method [30,31,32]. The bulk density of 3DVAG prepared by the KOH−assisted hydrothermal method is low, and it has limitations when loading active substances. This can only be compounded by subsequent in situ loading. The CVD method can be used to prepare high−quality 3DVAG with microscopic vertical channels, but it requires high temperature and high vacuum. The directional freezing method uses simple ice crystal templates to assemble graphene oxide (GO) flakes to obtain micron−scale vertically ordered graphene arrays with good orderliness, whose pore sizes and wall thicknesses can be controlled by adjusting certain processing parameters. The formation of vertical porous structures by the directional freezing method is affected by the complex dynamic liquid–particle and particle–particle interactions. During directional freezing, graphene oxides with a large size of area (LGO) are superior in promoting the formation of a good vertical porous structure, which has low interlayer contact resistance, good stress transfer efficiency, and excellent mechanical and electrical properties [33]. 

Herein, a hydrothermally assisted unidirectional freezing and sequential thermal reduction process was used to prepare 3DVAG with a vertically ordered honeycomb structure from LGO. When used in a three−electrode system, the prepared 3DVAG with a vertical structure showed higher specific capacitance, faster ion diffusion, and better magnification performance compared with three-dimensional reduced graphene oxide (3DrGO) of random structure. When the 3DVAG was constructed into a zinc−ion hybrid supercapacitor, excellent electrochemical performance was also obtained.

## 2. Experimental

### 2.1. Preparation of LGO

Flake graphite (200 mesh) was chemically oxidized and exfoliated to obtain large sheets of graphene oxide. Firstly, 28 g KMnO_4_ was added to 260 mL of H_2_SO_4_ at a temperature below 10 °C controlled by ice bath and stirred for 2 h. Subsequently, 6 g of flake graphite was added to the solution, which was then heated in a 35 °C water bath for 16 h. H_2_O_2_ was added until no bubbles were generated to obtain the primitive graphite oxide (PGO). The PGO solution proceeded to precipitation, removing the upper part and washing with water several times to obtain LGO.

### 2.2. Preparation of 3DVAG

The directional freezing method uses anisotropically grown ice crystals as a template to assemble dispersed graphene oxide sheets into a porous network with vertically oriented micropores. Figure 1 shows the preparation process of 3DVAG gel. Firstly, the LGO suspension (2 mL, 5 mg·mL^−1^) was evenly mixed with ascorbic acid (20 mg), and then the solution was heated in an oil bath for the first hydrothermal reduction to obtain a partially reduced graphene oxide (PrGO) hydrogel. Then, the PrGO was placed on the surface of the copper ingot impregnated with liquid nitrogen at −196 °C for unidirectional freezing. Then, a second oil bath was introduced to reinforce the structure, and deionized water was used to remove impurities. Finally, after freeze drying and reduction (at 800 °C), 3DVAG with a vertical channel was obtained. 

During the preparation process, the number of oxygen−containing groups in graphene oxide was controlled by adjusting the first hydrothermal time to enhance the π–π interaction between GO sheets. This allowed maintaining good dispersion and fluidity in the solution, such that ice crystals could adjust the assembly of GO flakes during the freezing process. Four hydrothermal times of 10, 20, 30, and 40 min were set at two oil bath temperatures of 90 and 100 °C. The samples obtained were named VAG90−10, VAG100−10, etc. The first number represented the oil bath temperature, while the second number represented the hydrothermal time.

### 2.3. Characterizations and Electrochemical Measurements

The surface morphology of the samples was characterized by a Quanta 200 Scanning electron microscope (SEM, The Dutch FEI). The chemical structure of LGO was characterized by Fourier−transform infrared (FTIR, Perkin Elmer Spectrum 100) spectroscopy, X−ray diffraction (XRD, Bruker D8−Advanced diffractometer, with Cu as target material; radiation wavelength λ = 0.15418 nm; scan rate of 5°·min^−1^ from 5° to 40°), and Raman spectroscopy (LabRAM HR Evolution).

A three−electrode system was used to conduct electrochemical tests on the prepared 3DVAG and conventional graphene gel 3DrGO materials. In 2 M ZnSO_4_ aqueous electrolyte, 3DVAG or 3DrGO material was used as the working electrode, a platinum sheet was used as the counter electrode, and Ag/AgCl was used as the reference electrode. The electrochemical performance was tested within a voltage window of 0–0.8 V. The carbon−based aqueous zinc−ion hybrid supercapacitor was assembled using zinc flakes as the negative electrode, 3DVAG material as the positive electrode, and 2 M ZnSO_4_ aqueous solution as the electrolyte for a two−electrode test. The electrochemical performance was tested by electrochemical impedance spectroscopy (EIS), cyclic voltammetry (CV), and galvanostatic charge–discharge (GCD). 

The specific capacitance (C_m_, F·g^−1^) was calculated using the following equation:(1)$$C_{m}=\frac{I\times \Delta t}{\Delta U\times m}$$
where I (A) was the constant discharge current, Δt (s) was the discharge time, ΔU (V) was the discharge voltage window (minus the voltage drop V_drop_), and m (g) was the mass of active material.

The energy density (E_m_, W·h·kg^−1^) was calculated using the following formula:(2)$$E_{m}=\frac{1}{2}\times \frac{C_{m}\times (\Delta U{)}^{2}}{3.6}.$$

The power density (P_m_, W·kg^−1^) was calculated using the following formula:(3)$$P_{m}=3600\times \frac{E_{m}}{\Delta t}.$$

## 3. Results and Discussion

### 3.1. Morphology and Structure of LGO

Exfoliated LGO flakes were characterized by SEM measurement as shown in Figure 2. It can be seen that the average size of the flake was about 50 μm, and the largest flake could even reach more than 100 μm.

The chemical composition and structure of LGO were characterized as shown in Figure 3. In the FTIR spectrum (Figure 3a), the wide absorption peak at 3405 cm^−1^ was due to tensile vibration of the hydroxyl O−H group, the absorption peak at 1735 cm^−1^ was caused by tensile vibration of the C=O group on the base plane of the GO flake, the 1630 cm^−1^ absorption peak corresponded to tensile vibration of the carboxylic acid group COOH, and the absorption peaks at 1254 and 1076 cm^−1^ corresponded to tensile vibration of the C−OH and C−O−C oxygen−containing groups, respectively [34,35]. There were abundant oxygen−containing functional groups on the surface of LGO, indicating that flake graphite was successfully oxidized to graphene oxide.

According to XRD diffraction (Figure 3b), there was a sharp characteristic peak at 2θ = 10.6°, corresponding to LGO on the (001) crystal plane. The layer spacing of LGO calculated by the Bragg equation (2dsinθ = nλ) was d = 0.834 nm, which is significantly larger than that of graphite (0.335 nm, 2θ = 26.5°). This shows that, during the graphite oxidation process, a large number of oxygen−containing functional groups were inserted between the layers, and the π−π interactions within the flake were destroyed, thereby expanding the interlayer spacing of the graphite to form graphite oxide.

Figure 3c shows the Raman spectrum of LGO. There were two characteristic peaks at 1350 and 1590 cm^−1^, corresponding to the D peak and G peak of carbon material, respectively. The D peak represents the absorption peak generated by the vibration of sp^2^ hybridized carbon atoms, reflecting the structural defects inside the carbon material. The G peak is caused by the in−plane vibration of sp^2^ hybridized carbon atoms, representing the orderliness of the graphite structure. Therefore, the strength of the D peak and G peak (I_D_/I_G_) is closely related to the defects and crystallinity of carbon materials [36,37]. The I_D_/I_G_ ratio of LGO was calculated to be 0.97, indicating that a large number of functional groups and structural defects were introduced between the graphite flakes during the oxidation process, which led to increased structural disorder of GO and decreased crystallinity.

### 3.2. Morphology and Structure of 3DVAG

The graphene gel prepared at different oil bath temperatures and hydrothermal times was characterized by SEM, and the results are shown in Figure 4. In the first hydrothermal reduction process, the volume of the gel gradually decreased with the increase in water heating time. When hydrothermally heated at 90 °C for 10−20 min, there were no obvious vertical through−holes in the cross−section of the gel (Figure 4a,b), while the volume decreased obviously when the oil bath lasted 40 min at 100 °C, and the gel inside was seriously crosslinked. This shows that an insufficient reduction time resulted in large π−π interactions between graphene oxide sheets, which was a disadvantage to the accumulation of the sheets during the unidirectional freezing process; thus, the ice crystals formed vertical channels. However, in the subsequent second hydrothermal process, the secondary reduction structure shrank and the flakes were restacked, resulting in a messy arrangement of the internal structure of the gel. This was also the case with VAG100−10 (Figure 4e). 

In the gel cross−section of VAG90−30 (Figure 4c), a good vertical orientation can clearly be seen. The partially reduced graphene oxide was rejected by the anisotropic ice crystals during the unidirectional freezing process and was stacked between ice crystals. Because partial reduction enhanced the π−π interaction between PrGO layers, the formed three-dimensional network was very stable and could maintain its structural integrity during thawing, which was not damaged in the subsequent secondary reduction process. This result was similar for VAG100−20 (Figure 4f).

Under the condition of 90 °C hydrothermal reduction for 40 min or 100 °C hydrothermal reduction for more than 30 min, the gel was crosslinked during the first hydrothermal process due to the long reduction time. The ice crystal template could not adjust the orientation vector of the GO layer during the freezing process, and ice crystals grew across the flake, thus preventing vertical pores from being formed inside the gel (Figure 4d,g,h).

From the above results, it can be found that the directional structure appeared at both 90 °C and 100 °C oil bath temperatures. Compared with other times, the vertical structure of the gel obtained by hydrothermal treatment for 30 min at 90 °C was more obvious. However, when the temperature was 100 °C, a similar structure could be obtained merely by reduction for 20 min, indicating that a higher temperature reduced the partial reduction time required. In addition, a higher hydrothermal temperature led to more obvious shrinkage of the gel volume. In the actual preparation process, the hydrothermal reduction reaction at 100 °C was fast, and a large number of oxygen−containing functional groups were decomposed into gas. Because the solvent of the GO suspension is water, it evaporates quickly at high temperature, which leads to pores forming in the gel, as shown in Figure 5. The inner surface of the pores was smooth and flat, which blocked the growth of ice crystals during the unidirectional freezing process; the resulting gel structure had low mechanical strength and was prone to breakage. Therefore, the 100 °C oil bath was not used for hydrothermal reduction in subsequent studies.

In order to further determine the hydrothermal reduction time in the oil bath at 90 °C, three times (25, 30, and 35 min) were set to explore the best first hydrothermal reduction time. As shown in Figure 6, there was partial orientation formed by ice crystal growth inside VAG90−25, but the overall structural strength was low, and structural damage was prone to occur during the SEM section preparation process (Figure 6a). Inside VAG90−30, there was still a good layer orientation (Figure 6b). A disordered and macroporous structure appeared inside VAG90−35 (Figure 6c). The long hydrothermal time led to crosslinking of the gel sheet, which was not conducive to the growth of ice crystals and hindered vertical orientation. However, compared with the conventional hydrothermal graphene gel (Figure 6d), its structure still had a certain degree of order, indicating that the ice crystal template can be oriented to assemble graphene flakes by controlling the degree of reduction of graphene oxide sheets. Some studies have found that, under certain freezing conditions, the PrGO dispersion after being reduced by ascorbic acid for 30 min (corresponding to a carbon content of 58.48 wt.% or a C/O ratio of 1.93) can form a honeycomb−like vertical orientation structure by unidirectional freezing [38]. In order to show the vertical pores in the gel structure of VAG90−30, its transverse and longitudinal sections were characterized by SEM. From the cross−sectional SEM (Figure 6e), it can be found that the pore size was about 20−30 μm. From the longitudinal section (Figure 6f), an obvious microscopic orientation can be observed, with ordered vertical channels, where the pore length could reach hundreds of microns. From the above results, it can be confirmed that, under the condition of a 90 °C oil bath, the PrGO obtained following the first hydrothermal reduction time of 30 min can be subjected to unidirectional freezing to obtain a highly vertical and orderly 3DVAG material.

In order to evaluate the hydrothermal reduction and thermal reduction of VAG90−30 graphene gel, the samples were tested by FTIR spectroscopy, XRD, and Raman spectroscopy. In Figure 7a, it can be found that, after ascorbic acid reduction and heat treatment at 800 °C, the absorption peaks of oxygen−containing groups at 1735, 1630, 1254, and 1076 cm^−1^ were significantly weakened. In addition, a new broad peak appeared at 1557 cm^−1^, related to stretching vibration of the C=C bond in the reduced graphene oxide [34], indicating that LGO was successfully reduced. It can also be found from the XRD image (Figure 7b) that, as the degree of reduction increased, the (002) diffraction peak at 2θ = 26.3° finally appeared in 3DVAG. According to the Bragg equation, a graphene layer spacing of d = 0.336 nm inside 3DVAG was calculated, indicating that the reduction degree of 3DVAG was high. The Raman characterization and analysis of the defects of the graphene sheet during the reduction process are shown in Figure 7c. LGO, PrGO, and 3DVAG all had obvious absorption peaks at 1347 cm^−1^ (D peak) and 1596 cm^−1^ (G peak). After reduction, the I_D_/I_G_ ratios of PrGO and 3DVAG were 1.03 and 1.36, respectively. The increase in I_D_/I_G_ indicates that, during the reduction process, the releases of gas during the decomposition of oxygen−containing functional groups destroyed the integrity of the flake, resulting in an increase in defects and disorder of graphene.

### 3.3. Electrochemical Performance

Figure 8a shows the CV curves of 3DrGO and 3DVAG at a large scan rate of 100 mV·s^−1^. It can be found that the CV curves of the two materials all presented a rectangular shape. This shows that graphene formed a surface electric double layer with electrostatic adsorption to achieve electrochemical energy storage during the zinc−ion storage process, which is a typical electric double−layer capacitance behavior. Specifically, 3DVAG exhibited a more ideal rectangular shape and a larger CV area, indicating that 3DVAG had ideal electrochemical performance and good specific capacitance during charging and discharging. It can be found from the GCD curves of the two materials (Figure 8b) that, at a current density of 0.5 A·g^−1^, the specific capacity of 3DVAG with a vertical channel was 66.6 F·g^−1^, while the specific capacity of 3DrGO with a random structure was only 59.9 F·g^−1^, the GCD curve of 3DVAG exhibited a symmetrical isosceles triangle, with ideal double−layer capacitance behavior. It can be seen from the two EIS (Figure 8c) images that 3DVAG exhibited low electrolyte ion diffusion resistance due to its ordered vertical channels, while 3DrGO had poor ion diffusion resistance due to the random stacking of internal graphene sheets. The above results show that the vertical structure of 3DVAG exhibited better electric double−layer capacitance performance and ion diffusion rate than the random structure of 3DrGO material, thus providing structural advantages for the efficient storage of zinc ions. 

Figure 9a shows the CV curve of 3DVAG material at a scan rate of 5−100 mV·s^−1^. At different scan speeds, the CV curve could maintain a good rectangular shape, indicating good magnification performance. From the GCD curve shown in Figure 9b, it can also be found that the curves showed symmetry at different current densities. According to the specific capacitance calculation formula, it can be concluded that, as the current density increased, the specific capacitance did not have a large attenuation. The current density increased from 0.5 to 2 A·g^−1^, the specific capacitance decreased from 66.6 F·g^−1^ to 61.4 F·g^−1^, and the capacitance retention rate reached 92.2%. However, the conventional graphene gel 3DrGO showed poor rate performance (Figure 9c) at a high current density of 2 A·g^−1^, while its specific capacitance was only 78.4% at 0.5 A·g^−1^. This result shows that 3DVAG had excellent rate performance.

As shown in Figure 10a, a carbon−based aqueous zinc−ion hybrid supercapacitor was constructed using zinc flakes as the negative electrode, 3DVAG material as the positive electrode, and 2 M ZnSO_4_ aqueous solution as the electrolyte. Figure 10b shows the CV curve of the ZHSCs at a scan rate of 5 to 100 mV·s^−1^ and a voltage window of 0.2 to 1.7 V. Its shape was rectangular, showing that the 3DVAG positive electrode stored zinc ions through the pore absorption/desorption reaction, and the negative electrode featured the deposition/dissolution reaction of zinc ions on the surface of the zinc sheet. With the gradual increase in scanning speed, the CV response current and the area became gradually larger, and the shape did not undergo obvious deformation. Figure 10c shows the GCD curve of ZHSCs at different current densities of 0.3−2 A·g^−1^. The curve showed a certain voltage plateau, corresponding to the redox peak in the CV curve, thus exhibiting good zinc−ion storage performance. Through calculation, the mass specific capacitances at current densities of 0.3, 0.5, 1, 1.5, and 2 A·g^−1^ were 44.4, 33.2, 29.1, 26.7, and 25.5 F·g^−1^, respectively, showing a good rate performance (Figure 10d). In addition, the ZHSCs also showed excellent cycle stability, whereby the capacitance retention rate reached 94.6% (Figure 10e) after 3000 cycles of charge and discharge at a current density of 2 A·g^−1^. Figure 10f shows the Ragone diagram of 3DVAG//Zn ZHSCs, it can be found that the 3DVAG electrode exhibited high energy and power density. At a power density of 249.3 W·kg^−1^, its energy density could reach 17.03 W·h·kg^−1^; at a high power density of 1528 W·kg^−1^, its energy density could reach 8.28 W·h·kg^−1^, which is higher than some previously reported carbon−based supercapacitors [39,40].

## 4. Conclusions

In this study, 3DVAG material was prepared by the hydrothermal−assisted unidirectional freezing method, and its electrochemical performance was investigated for ZHSCs. The structure of three-dimensional graphene can be controlled by the temperature and time of the hydrothermal reduction. Compared with 3DrGO, 3DVAG material exhibited higher specific capacitance, faster ion diffusion, and better rate performance. ZHSCs constructed with 3DVAG materials exhibited a wide voltage window, excellent cycle stability, and high energy density. Thus, 3DVAG provides a structural advantage for the selection of electrode materials used in capacitors to achieve efficient and stable electrochemical energy storage.

## Figures and Tables

**Figure 1 molecules-27-00376-f001:**
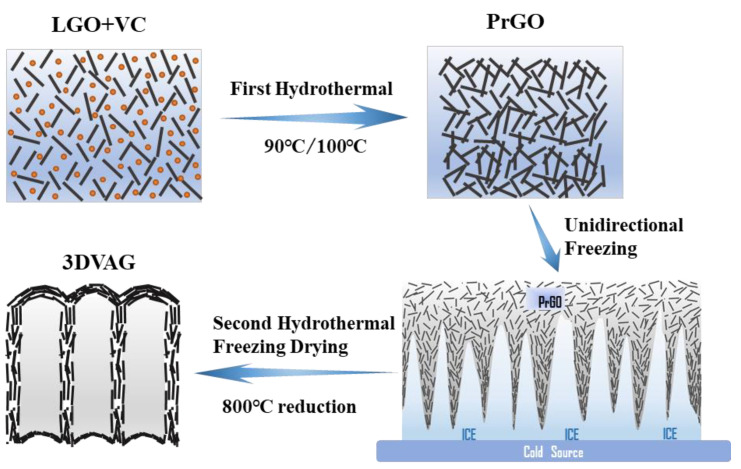
Preparation flowchart of 3DVAG.

**Figure 2 molecules-27-00376-f002:**
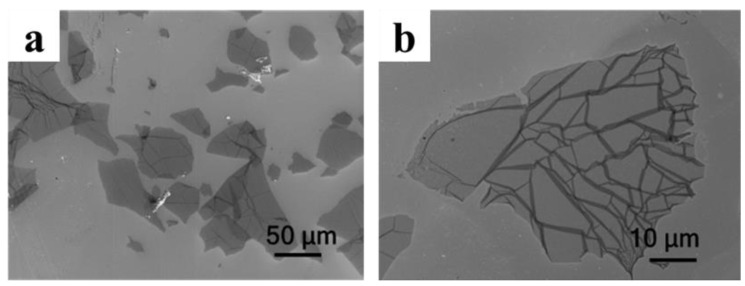
SEM image of LGO. (**a**) SEM image with 50μm scale; (**b**) SEM image with 10 μm scale.

**Figure 3 molecules-27-00376-f003:**
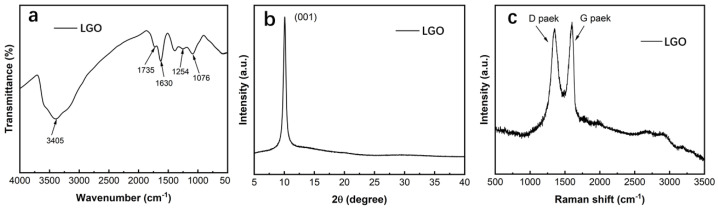
Chemical composition and structural characterization of LGO: (**a**) FTIR spectrum; (**b**) XRD; (**c**) Raman spectrum.

**Figure 4 molecules-27-00376-f004:**
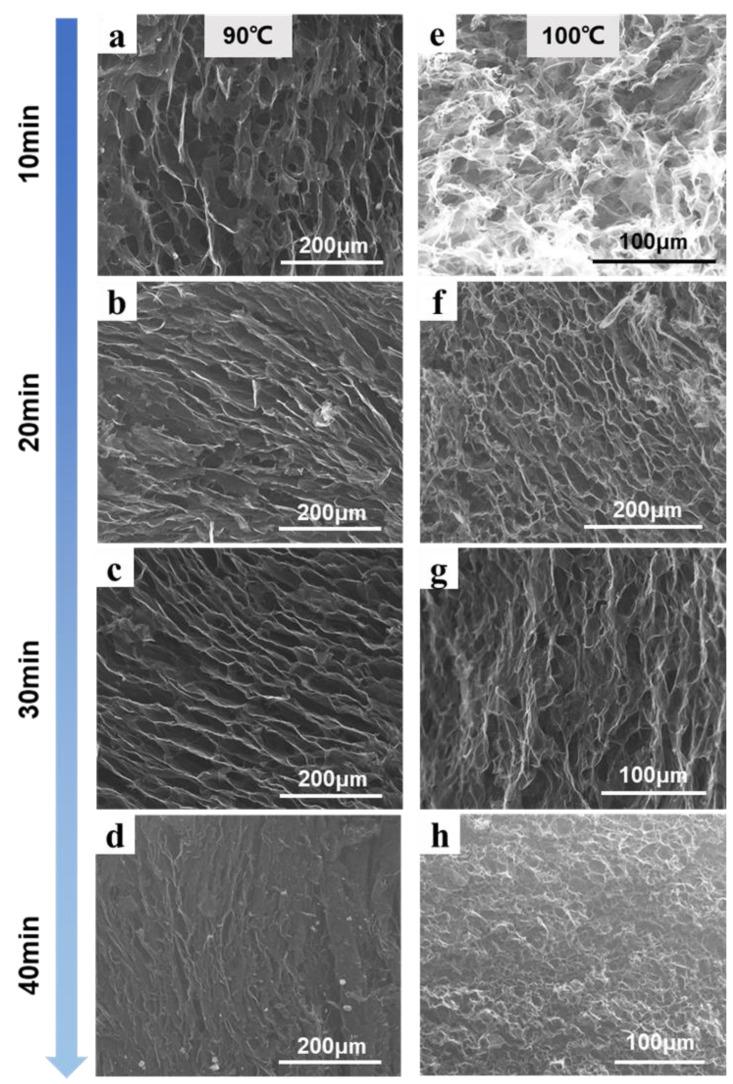
Effect of hydrothermal temperature and reduction time on the microstructure of graphene gel. (**a**) hydrothermal reduction at 90 °C for 10 min; (**b**) hydrothermal reduction at 90 °C for 20 min; (**c**) hydrothermal reduction at 90 °C for 30 min; (**d**) hydrothermal reduction at 90 °C for 40 min; (**e**) hydrothermal reduction at 100 °C for 10 min;(**f**) hydrothermal reduction at 100 °C for 20 min; (**g**) hydrothermal reduction at 100 °C for 30 min; (**h**) hydrothermal reduction at 100 °C for 40 min.

**Figure 5 molecules-27-00376-f005:**
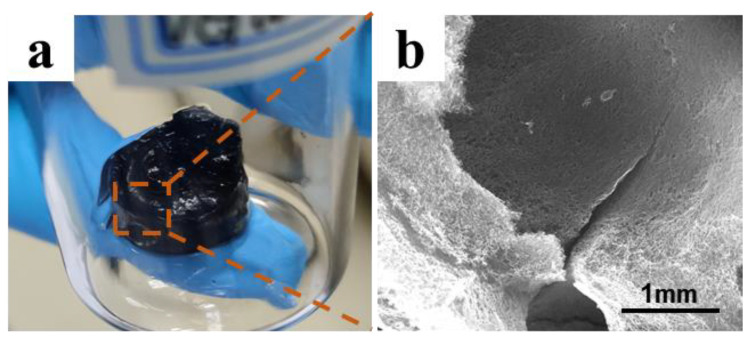
Pore and cracks in the gel after hydrothermal reduction at 100 °C: (**a**) surface of the gel; (**b**) inner surface of the pores.

**Figure 6 molecules-27-00376-f006:**
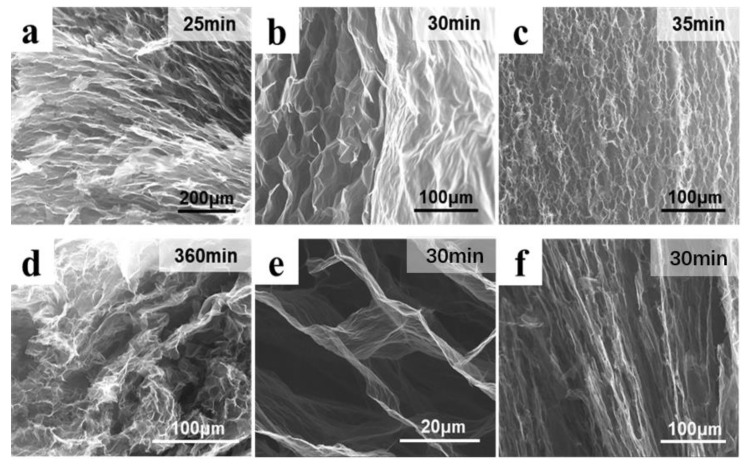
The micromorphology of graphene gel prepared in diffident conditions: (**a**) hydrothermal reduction of 25 min; (**b**) hydrothermal reduction of 30 min; (**c**) hydrothermal reduction of 35 min; (**d**) hydrothermal reduction of 360 min; (**e**) cross section of VAG90−30; (**f**) longitudinal section of VAG90−30.

**Figure 7 molecules-27-00376-f007:**
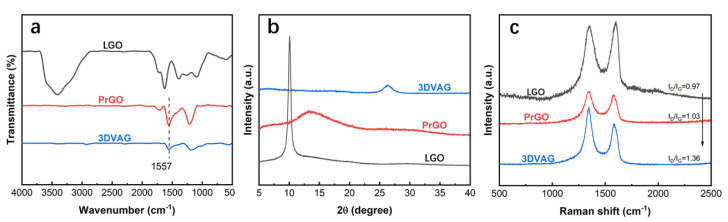
Chemical composition and structural characterization of different materials: (**a**) FTIR spectrum; (**b**) XRD; (**c**) Raman spectrum.

**Figure 8 molecules-27-00376-f008:**
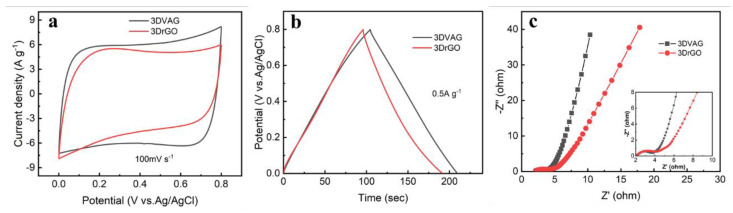
Electrochemical performance of 3DVAG and 3DrGO in a three−electrode system: (**a**) CV; (**b**) GCD; (**c**) EIS.

**Figure 9 molecules-27-00376-f009:**
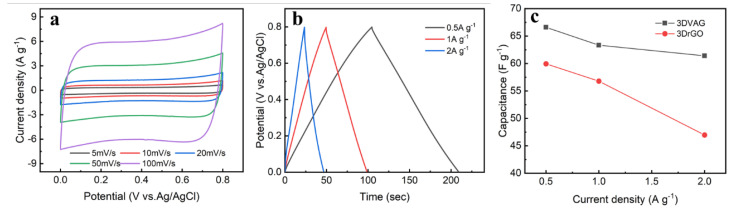
Electrochemical performance of 3DVAG material under a three−electrode system: (**a**) CV curves at different scan rates; (**b**) GCD curves at different current densities; (**c**) comparison of rate performance with 3DrGO.

**Figure 10 molecules-27-00376-f010:**
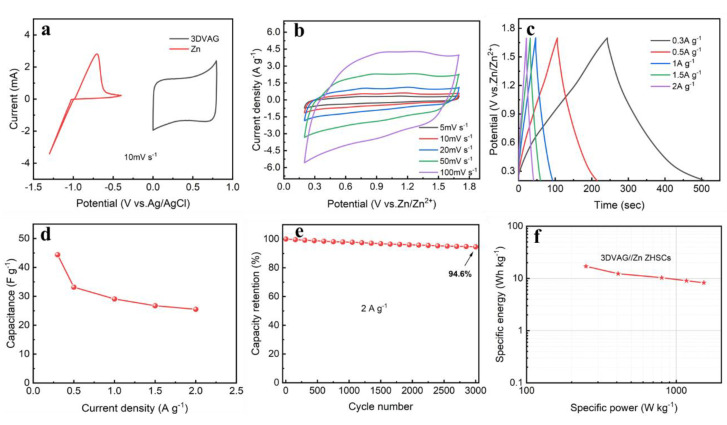
Electrochemical performance of ZHSCs constructed by 3DVAG materials: (**a**) carbon−based aqueous zinc ion hybrid supercapacitor; (**b**) CV curve; (**c**) GCD curves at different current densities; (**d**) rate performance; (**e**) cycle life; (**f**) Ragone diagram.

## Data Availability

The data presented in this study are available from the corresponding author upon reasonable request.

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
