# Peer review of "Preparation and Electrochemical Performance of Three-Dimensional Vertically Aligned Graphene by Unidirectional Freezing Method"

_molecules, 2022, doi:10.3390/molecules27020376_

Round 1

Reviewer 1 Report

1. Please check the plagiarism

2. Grammatical corrections are needed

3. Rewrite figure captions properly. Fig. 1, 2, 4, 6

4. Why CV and GCD of the device were measured not from E = 0 V.  Justify?

5.  Explain the CV of Zn flakes electrode used as anode for SC device, Figure  10 a, whether Y-axis is mA cm-2 or mA g-1, Clarify?--> How calculations of energy density (Whkg-1)and power density (W kg-1)of the device were performed?

Reviewer 2 Report

The manuscript ID: molecules-1528616, entitled “Preparation and Electrochemical Performance of Three-Dimensional Vertically Aligned Graphene by Unidirectional Freezing Method” was carefully reviewed. The manuscript reports electrochemical performances of three-dimensional vertically aligned graphene (3DVAG) that is prepared by a unidirectional freezing method. The investigation method is interesting and there is a good connection between sections. However, the manuscript needs further improvement, and I recommend the publication of this manuscript after moderate revision and modification based on the provided comments.

1. Authors need to mention advantages and disadvantages of this method (Unidirectional Freezing Method) compared to the well-known Chemical Vapor Deposition (CVD) process in the introduction section. In addition, the below article that covers all aspects and advantage of CVD process compared to others need to be cited and mentioned in the introduction section:

“L. Sun, G. Yuan, L. Gao, J. E. Yang, M. Chhowalla, M. Heydari Gharahcheshmeh, K. Gleason, Y. S. Choi, B. H. Hong, Z. Liu, Chemical Vapor Deposition, Nature Reviews Methods Primers, 1, 5, (2021).”

2. Discussion related to the figure 4 (effect of temperature and reduction time) need be expanded further with more details and scientific aspects.

3. Authors need to report the process parameters information of the sample in Figure 3b. Do different process parameters influence the lattice parameter?

4. The time scale for figure 6-e and figure 6-f is missing from the figure and please insert that information in the figure.
